# The Quality of Life and Associated Factors Among Older Adults in Central Nepal: A Cross-Sectional Study Using the WHOQOL-OLD Tool

**DOI:** 10.3390/ijerph22050693

**Published:** 2025-04-27

**Authors:** Rubisha Adhikari, Rajani Shah, Kamal Ghimire, Birat Khanal, Sunil Baral, Anisha Adhikari, Dinesh Kumar Malla, Vishnu Khanal

**Affiliations:** 1Shree Medical and Technical College, Purbanchal University, Bharatpur 44207, Nepal; adhikarirubisha@gmail.com; 2School of Health Science, Council for Technical Education and Vocational Training (CTEVT), Bharatpur 44200, Nepal; 3Bloomberg School of Public Health, Johns Hopkins University, Baltimore, MD 21205, USA; 4Universal College of Medical Science, Bhairahawa 32900, Nepal; 5Chitwan Medical College, Bharatput 44200, Nepal; sunil_baral2008@icloud.com; 6Manmohan Memorial Institute of Health Sciences, Tribhuvan University, Kathmandu 44614, Nepal; 7Birendra Multiple Campus, Tribhuvan University, Bharatpur 44207, Nepal; 8Menzies School of Health Research, Charles Darwin University, Remote Health Systems and Climate Change Centre, Alice Springs, NT 0870, Australia

**Keywords:** older people, quality of life, urban health, Nepal, WHOQOL-OLD

## Abstract

Ensuring people’s quality of life (QOL) has become increasingly challenging due to population aging. This study aimed to investigate the QOL among older people and factors associated with it in an urban setting of Central Nepal using the World Health Organization Quality of Life (WHOQOL-OLD) tool. A cross-sectional study was conducted in Central Nepal. The association between QOL and independent variables was first examined using a univariate analysis of variance followed by multiple linear regressions. The mean age of the 366 participants was 70 years (standard deviation [SD]: 8.2 years). The mean of the overall QOL scores was 74.37 (SD: 7.82). Older people who were literate (regression coefficient (β): 1.909; 95% confidence interval (CI): 3.771 (1.986, 5.556)), who had an annual household income of NPR 40,000 (Nepalese Rupees) or more (β: 1.909: 95% CI: 0.337, 3.480), who reported health services as accessible (β: 4.019; 95% CI: 0.666, 7.371) and affordable (β: 3.176; 95% CI: 1.327, 5.025), and who reported partaking in physical activity (β: 2.107; 95% CI: 0.607, 3.606) had higher QOL scores compared to their respective counterparts. A holistic model of service using the social determinants of health framework is essential to improve the well-being of older people in Nepal.

## 1. Introduction

The majority of individuals worldwide can now anticipate living into their 60s and beyond [1]. The additional years of life and changes in the population have significant impacts on each of us individually and on society [2]. Various factors such as genetics, lifestyle choices, eating a nutritious diet, quitting smoking, and engaging in physical activity have contributed to this increased life expectancy. They present previously unattainable opportunities and are likely to fundamentally alter the way we live, what we strive for, and how we interact with one another [3]. Over the coming decades, population aging will be a global phenomenon experienced by all regions and places [4]. Low- and-middle-income countries (LMICs), however, face significant pressure from the aging population since their social security, health, and geriatric systems are still in their infancy [5].

Older people are considered as repositories of knowledge, life experiences, and collections of various ideas. Their knowledge, wisdom, and conscience can be beneficial to national progress and prosperity [6]. In 2019, there were 703 million older people worldwide who were 65 years of age or older [7,8]. Health issues affecting individuals aged 60 years or above account for 23% of the world’s overall disease burden, according to Global Burden of Disease (GBD) estimates for 2010 [9]. Cardiovascular illnesses share the largest portion of the disease burden in individuals 60 years of age and older (30.3%), followed by malignant neoplasms (15.1%), chronic respiratory diseases (9.5%), musculoskeletal diseases (7.5%), and neurological and mental disorders (6.6%) [9]. With the rising older population, the healthcare demand will increase substantially [10]. The health services in LMICs have not been ready to adjust to the growing needs of an aging population. Moreover, their focus has mainly been on managing disease burden, such as of communicable diseases, child mortality, and maternal mortality [11].

According to the World Health Organization (WHO) “A person’s quality of life (QOL) refers to how they feel about themselves within their own culture and value system” [12]. Quality of life also deals with the goals, standards, concern, and expectations of life of individuals [13]. Furthermore, the physical, functional, social, and emotional elements that contribute to an individual’s health all have an impact on their QOL [14]. Various factors such as education, income, access to health services, healthcare affordability, co-morbidities, physical activities, and participation in social activities have been found to be associated with QOL. For instance, education contributes to better health literacy and increases one’s capacity to navigate health systems [15]. Better household income is linked to better nutrition, living conditions, and the ability to pay for healthcare, all of which enhance the quality of life later in life [16]. Additionally, it has been demonstrated that regular exercise helps elderly populations preserve their mobility, lower their risk of non-communicable diseases, and improve their mental health [17]. The influence of these factors on QOL are multifaceted and thus highlights the complexity of healthy aging [18]. Understanding the QOL of older people is an important first step to identify vulnerable subgroups within the older population and necessary actions to improve their well-being.

The life expectancy in Nepal has increased from 28 years in 1952 to 72 years in 2021 [19]. An older person is defined as someone aged 60 years or above [20], and recent estimates from the 2021 census suggest that there are 2.97 million older people in the country [21]. Between 2011 and 2021, there was an increase in the older population by 2.07% in Nepal [22]. There is a high rate of illiteracy poverty, overuse of land resources, stagnating economy, and poor health in the country, and even a slight increase in the proportion of older people would raise concern [6]. The Government of Nepal (GoN) has developed the “National Health Policy 2017” to meet the demands of the country’s population [21]. A few initiatives, such as the free Health Care Service Program for older people, “*Jeshtha Nagarik Swashthopachar Kosh*” (Senior Citizens Health Facilities Fund), and Old-Age-Allowance, have been implemented in recent years to address the healthcare needs of older people into practice [23].

Only a few studies have reported on the QOL of older people in Nepal. Some of these have focused on participants with identified health conditions, such as those with HIV/AIDS and receiving support and treatment [24,25] or those confined to hospital settings in major cities [26] who were attending an outpatient clinic in a Kathmandu-based hospital, and was focused on life satisfaction rather than QOL. It should be noted that the study location, Kathmandu, is very different from other cities, towns, and villages in Nepal. Kathmandu is the capital city, where most of the country’s resources have been poured to build health and social infrastructure, making it quite an affluent location to live. The well-being of older adults in Kathmandu can be positively impacted by their greater access to social services, specialist healthcare, and transportation [27]. In comparison, the current study setting, Chitwan, is a developing metropolitan city, and lacks community-based senior-focused programs and geriatric healthcare facilities, all of which can have detrimental impacts on the aging population’s QOL [28]. It is also noteworthy that studies conducted in clinical settings, or among participants with certain health issues, do not reflect the status of the general older population. Another study was solely focused on rural citizens [29] of the far western region of Nepal, which is socially, culturally, and geographically different from other regions of Nepal. Although Nepal is relatively small, it is incredibly diverse in its geography, and access to social and healthcare and infrastructure. Therefore, it is essential to understand the unique situations in various settings of Nepal to develop locally relevant policies and programs [30].

The World Health Organization’s Quality of Life-Old (WHOQOL-OLD) was developed especially for older people, has undergone cross-cultural psychometric validation, and ensures accurate assessment in important areas including independence, social interaction, and sensory perception [31]. The tool helps collect and report QOL data to ensure consistency and comparability globally. Such standardization and comparability are essential for policymakers to design appropriate policies and programs targeted at the older population. There is a paucity of literature on the QOL of community-based older participants in urban settings of Nepal using a WHO-developed and -validated tool [13]. Therefore, this study aims to determine the QOL of the older person and associated factors in an urban setting of Central Nepal using the WHOQOL-OLD tool.

## 2. Materials and Methods

### 2.1. Study Context

This study was conducted in Bharatpur Metropolitan City, the headquarter of Chitwan district, situated in the central-southern part of Nepal [32]. The total population of the city was 369,377 in 2021, of which 29,413 were 60 years or older [33]. The majority of the healthcare facilities are run by private for-profit services, although public hospitals and teaching hospitals also offer free health services. For older citizens, the GoN provides free medical care at government health facilities, along with financial assistance for a limited number of diseases [23]. Such services are delivered via public facilities only [23]. However, only very few older people are aware of the free healthcare and subsidy programs [34]. There are no mandatory state-supported retirement funds, which leaves the majority of seniors dependent on their family or children for financial support [20].

### 2.2. Study Design and Participant Selection

This community-based cross-sectional study was conducted between April and May 2023. The sample size for this study was 310, using the following assumptions: an average 82.41% prevalence of fair QOL [35], a 6% margin of error and a non-response rate of 10% [36]. We inflated the sample size (N = 366) to account for any missing values.

In total, 5 of the 29 wards of Bharatpur Metropolitan City were selected randomly for this study. We included older people aged 60 years or above of both genders who gave informed consent to participate. We excluded those who could not respond to the questionnaire due to their illness, those with cognitive impairment, or those who could not communicate. A list of the households was obtained from the local municipality office, and a population proportionate to size method was used to recruit the required number of participants. Where there were two or more older people in one household, one older person was chosen randomly for participation.

### 2.3. Instrument and Data Collection

This study used the WHOQOL-OLD tool to measure the QOL of older people. This is a cross-culturally reliable, 24-item, 6-facet measure developed by the WHO expert group [37]. This tool was found to be valid for the Nepalese older population in rural Nepal [35] and was subsequently used in an urban area [38]. The English version of the tool was translated to Nepali language (RA and AA) and checked by two senior researchers (RS and DM). The Nepali version was pretested among 30 older people residing in other wards of the city. Some minor modifications were made after pretesting, including re-ordering the questions and modifying some words to reflect the local dialect and structure, which would make it easier for the participants to understand. The final version of the questionnaire included questions on socio-economic and demographic factors, the role of family support, chronic conditions, physical activity, and health services. Face-to-face interviews were conducted with participants after obtaining consent.

### 2.4. Variables

#### 2.4.1. Outcome Variables

The outcome of this study was QOL score, measured using a five-point Likert scale based on 24 items from the WHOQOL-OLD instrument. This tool comprises six domains: sensory abilities; autonomy; past, present, and future activities; social participation; death and dying; and intimacy [12]. Each domain contains 4 items, with responses rated from 1 to 5. Each domain’s scores range from 4 to 20. The overall QOL score ranges from a minimum of 24 to a maximum of 120 [39]. The QOL was calculated by averaging the scores of the individual domains and the overall score of all six domains and was reported as continuous variables.

#### 2.4.2. Independent Variables

Independent variables were selected based on previously published studies [20,26,35,40]. Sociodemographic variables included age, gender, ethnicity, religion, marital status, family type, living status, education, household income, employment status, and personal income. Age and monthly household income were recorded as continuous variables. Age was then recoded into two categories: (i) 60–69 and (ii) ≥70, based on the median age (70 years) as the cutoff. Likewise, monthly household income (in Nepalese Rupees) was categorized into a dichotomous variable: (i) below NPR 40,000 and (ii) NPR 40,000 and above, based on the median value of NPR 40,000. Gender was recorded as (i) male or (ii) female. Ethnicity was categorized based on Nepal’s caste system [26]. Despite being legally abolished, Nepal’s caste system still exists and has Brahmins and Chhetris at the upper end and Dalits at the lower end of the spectrum, which continues to affect social hierarchies and access to resources [41]. For our study, we categorized castes/ethnicity into upper caste, Janajati (indigenous), and Dalit (lowest hierarchy in the caste system). Religion was dichotomized into (i) Hindu and (ii) other (Muslim and Buddhist) to facilitate analysis. Marital status was recorded as (i) married or (ii) all others (unmarried/separated/widow/widower); family type as (i) nuclear or (ii) joint family; and living status as (i) living with son, (ii) living with spouse, or (iii) other (daughter/alone/mother). Education status was recorded as (i) illiterate (those who could not read or write) and (ii) literate. Employment status was recorded as (i) employed or (ii) unemployed. Personal income was categorized as (i) yes, for those who had their own source of income, or (ii) no, for those not having this. Social security allowance was recorded as (i) yes, they receive the support, or (ii) no, for otherwise. Food security was assessed by asking participants whether they had adequate food over the past 12 months. The responses were collected as (i) enough food for less than six months, or (ii) enough food available and affordable for six months or more. Likewise, tobacco use was also recorded as (i) yes, if they had ever used, or (ii) no, if they had not. Current disease status was recorded as (i) yes, if they reported having any disease, or (ii) no, if they did not have any disease (self-reported) at the time of the study. The family support section consisted of two responses: (i) no, for those not receiving support from family, or (ii) yes, for those receiving support from family. Family support was further divided into daily work support, decisional support, economic support, and emotional support.

Three key aspects of health services such as accessibility, availability, and affordability were measured using three items based on the existing literature [35,42]. Accessibility, referring primarily to issues affected by distance, comprised three items with a total score between 0 and 3 points. If they scored 0–1, they were categorized as reporting ‘no accessibility’, and if 2–3, then they were categorized as reporting ‘accessibility’. Similarly, availability referred to services available, consisting of three items with three choices: 1, 2, and 3. The total score ranged from 1 to 9 points. It was categorized as ‘not available’ if score was <6 points and ‘available’ if score was ≥6 points. Lastly, affordability contained three items with a total score between 0 and 6 points, categorized as ‘affordable’ (5–6 points) and ‘not affordable’ (<5 points).

### 2.5. Statistical Analysis

Descriptive analyses were performed to present the frequency distribution as a count and percentage. Individual and overall mean scores (with standard deviation (SD)) were computed for all domains of QOL. The relationship among the individual domains of QOL was examined using Pearson’s correlation coefficient. We examined the distribution of the overall mean QOL score by different independent variables using *t*-test or analysis of variance (ANOVA) test, as appropriate. Variables that were significantly associated with overall QOL in *t*-test or ANOVA were further subjected to multiple linear regression to examine their association with overall QOL. We also repeated the regression models for the six individual domains of the QOL to investigate which factors remained consistently associated with the various domains. The assumption of normal distribution was tested using normality tests, Q-Q plots, and histograms. The main outcome of the study, i.e., ‘overall quality of life’ was normally distributed. A *p*-value of <0.05 was set as statistically significant. Data analysis was performed using Statistical Package for Social Science (SPSS) version 28 [43].

### 2.6. Ethics

The ethics approval for the study was obtained from the Institutional Review Committee (IRC) of Shree Medical and Technical College (SMTC-IRC−20230217−68). The study was explained to all the participants in Nepali and written informed consent was obtained prior to conducting the interviews. Participants were informed that they had the right not to participate or withdraw from the study at any time without any consequences. Only anonymized data were used for the analysis.

## 3. Results

### 3.1. Characteristics of the Study Population

Table 1 presents the characteristics of the 366 participants in this study. Briefly, the mean age of the participants was 70 years (standard deviation: 8.2 years). Of which, 56.3% were aged 70 years and above, 55.2% were female, and 81.7% were of the upper caste. A vast majority of them (78.1%) lived with a joint family. Only one-third were literate, 7.7% were employed, 44.5% had a household income < NPR 40,000 (Nepalese Rupees), 85.0% had no personal income, and 66.9% of the participants reported receiving a social security allowance.

### 3.2. QOL Among Older People

Table 2 shows the overall QOL scores with a mean of 74.37 (SD: 7.82). The mean score was found to be higher for the past, present, and future activities (14.96, SD: 2.32); social participation (14.29, SD: 2.17); and intimacy (14.69, SD: 2.36) compared to the other domains of QOL (Table 2). The correlation among the different domains of the QOL is shown in Appendix A. A statistically significant correlation was found between autonomy and the past, present, and future activity domains; social participation and the past, present, and future activity domains; death and dying and sensory ability domains; and intimacy and the other three domains (autonomy; past, present, and future activities; and social participation).

### 3.3. Factors Associated with QOL Among Older People

The results of the QOL score according to different sociodemographic variables are presented in Appendix A. The mean QOL score varied significantly by age, gender, religion, education, social security, alcohol consumption, emotional support, decisional support, economic support from family, availability, affordability, and physical activity in the *t*-test or ANOVA, which were subjected to multiple linear regression.

Table 3 presents the results of the multiple linear regression, which was performed to identify factors associated with the overall QOL. Older people who were literate (regression coefficient (β): 1.909: 95% confidence interval (CI): 3.771 (1.986, 5.556)), who had an annual household income of NPR 40,000 or more (β: 1.909: 95% CI: 0.337, 3.480), who reported health services as accessible (β: 4.019; 95% CI: 0.666, 7.371) and affordable (β: 3.176; 95% CI: 1.327, 5.025), and who reported partaking in a regular physical activity (β: 2.107; 95% CI: 0.607, 3.606) had higher QOL scores compared to their respective counterparts. Conversely, the participants who reported that health services were available (β: −2.011; 95% CI: −3.585, −0.436) had lower overall QOL scores compared to their counterparts. We performed a sensitivity analysis, treating the participants’ age and income as continuous variables in the multiple regression model. While education, household income, accessibility, availability, affordability, and physical activity remained significantly associated with the overall quality of life, the respondents’ age did not show a significant association. Table 4 presents additional analyses to examine the factors associated with the individual domains of QOL using multiple linear regression. Gender was significantly associated with the QOL domain of autonomy (*p* = 0.02) and death and dying (*p* < 0.01). The participants’ education was related to autonomy (*p* < 0.001); past, present, and future activities (*p* < 0.01); social participation (*p* = 0.01); and intimacy (*p* = 0.03). The household income also demonstrated a significant association with autonomy (*p* = 0.04); past, present, and future activities (*p* = 0.02); social participation (*p* = 0.01); and intimacy (*p* = 0.01). Food security was linked to past, present, and future activities (*p* = 0.03). Alcohol consumption was connected to autonomy (*p* = 0.05), social participation (*p* = 0.02), and intimacy (*p* = 0.06). The availability of health services showed a significant association with the domains of autonomy (*p* = 0.01); past, present, and future activities (*p* < 0.01); and intimacy (*p* = 0.02). Similarly, the affordability of health services was significantly associated with autonomy (*p* = 0.01); past, present, and future activities (*p* < 0.01); and social participation (*p* < 0.01). Regular physical activity was related to both autonomy (*p* = 0.01) and social participation (*p* = 0.01).

## 4. Discussion

This study examined the QOL of older people in relation to six measurable domains and their associated factors in an urban setting in Central Nepal. It showed that education, household income, health service accessibility and affordability, and physical activities were associated with the QOL. While there were considerable variations among the factors associated with each domain of QOL, the association of education, household income, physical activity, and health services remained consistent. To the best of our knowledge, this is one of the first studies to use the WHOQOL-OLD tool in urban Nepal; therefore, the knowledge generated from this study is unique.

Two key factors intertwined with health services, i.e., affordability, and accessibility, were associated with a higher QOL in this study. Similar findings were observed in studies conducted in Nepal and Bangladesh, indicating that the affordability of health services was positively associated with the QOL among older persons [35,44].

A higher household income was positively associated with quality of life, as older people with a household income of NPR 40,000 and above had higher QOL scores than others. Similar findings were observed in another study among older patients, where a high family income was positively correlated with quality of life; however, direct comparison cannot be made due to the methodological differences [26]. The household income determines the financial strain. Lower household incomes and financial strains have been reported to be associated with a lower QOL in Europe [45], although these findings need to be compared with caution due to differences in the research settings and context. The household income also reflects the ability of older people to afford basic services, healthcare services, and facilities to help them live more comfortably.

A unique finding we noted in this study is that the availability of health services was negatively associated with QOL. The reason behind this association remains unclear, and there are no other studies that have explained this aspect in Nepal or similar contexts. Further research is warranted to elucidate the mechanisms contributing to this counterintuitive finding.

Being physically active was also positively associated with QOL. Physical activities help improve general physical and mental health, improve muscle tone and mobility, and prevent falls among older adults. These factors individually as well as combined can help improve the overall QOL. For instance, in our data, those who reported partaking in physical activity had a mean QOL 75.85 (SD: 7.84) compared to those who reported no physical activity (mean: 72.84; SD: 7.38). Mobility is important in all contexts, even more so in low-resource settings such as Nepal, where assisted living is not available. A study among older Bangladeshi adults demonstrated that those who partook in moderate physical activities reported a better QOL [44]. A systematic review and meta-analysis of 12 studies among older adults also concluded that physical activities were the facilitators of a better QOL [46].

Our study showed that literacy was associated with a higher likelihood of a better QOL, consistent with results from earlier studies in Nepal [29,35,40]. Studies from Sri Lanka [47], Malaysia [48], and Vietnam [49] have also reported a similar finding between higher education and improved QOL. Those with higher education attainments are more likely to adopt healthy habits and have better access to healthcare compared to those with lower education levels, which could lead to a better QOL among educated older persons [50].

This study presents certain methodological limitations; however, it also showcases several significant strengths. The study tool was developed by the WHO expert group [13], ensuring its validity and reliability in our setting. We also included important variables related to health services, such as accessibility, availability, and affordability, which were not addressed in previous studies from Nepal. Key limitations of the study include its focus on only one urban area. While the findings are reflective of urban settings in Nepal, they may not be generalizable to all settings within the country. Although we included the most relevant variables in this study, there may be factors influencing QOL that were not captured, for instance, factors such as loneliness, nutrition status, relationship with children, receipt of support from the government, and mobility-related issues may have a significant bearing on QOL. Since the post-regression residuals for sensory ability and death and dying were not normally distributed, the results should be interpreted cautiously. The other domains of QOL and the overall QOL (primary outcome variable), however, had normally distributed post-regression residuals. Finally, the current study did not conduct validation prior to using the WHOQOL-OLD tool due to time limitations. Although the tool has been found to be valid previously in Nepal, such a validation exercise should be conducted to reflect the current social changes in the country. Nevertheless, this study addresses many important factors influencing the quality of life of older people in Nepal.

## 5. Conclusions

This community-based study provides insight into the attributes of quality of life of older persons in Nepal. As the older population is growing rapidly, maintaining and promoting a better quality of life has become a major priority and challenge. This study showed that key factors associated with QOL among older people were household income, literacy status, accessibility, affordability and availability of health services, and physical activity. While multiple domains of QOL were associated with education attainment and household income, it is important to appreciate that many older people may not be educated and have no financial means to support themselves. Therefore, local government-initiated programs are necessary to create a support network. The federal and local governments should create infrastructures and systems to help facilitate meaningful engagement of older people and play an important role in ensuring that health and social services are accessible to them. Future research needs to focus on the models and feasibility of contextualized interventions to improve the quality of life and well-being of older people in Nepal.

## Figures and Tables

**Table 1 ijerph-22-00693-t001:** Characteristics of participants (N = 366).

Variables	Frequency (%)
**Age (in years)**	(median: 70.00)
60–69	160 (43.7)
≥70	206 (56.3)
**Gender**	
Male	202 (55.2)
Female	164 (44.8)
**Ethnicity**	
Upper	299 (81.7)
Janajati	52 (14.2)
Dalit	15 (4.1)
**Religion**	
Hindu	329 (89.9)
Others *	37 (10.1)
**Type of family**	
Nuclear	80 (21.9)
Joint	286 (78.1)
**Living with**	
Son	294 (80.3)
Spouse	45 (12.3)
Others ^#^	27 (7.4)
**Marital status**	
Married	262 (71.6)
Unmarried/separated/widow/widower	104 (28.4)
**Education status**	
Illiterate	247 (67.5)
literate	119 (32.5)
**Employment status**	
Employed	28 (7.7)
Unemployed	338 (92.3)
**Household income** (monthly)	(median: 40,000)
<NPR 40,000	163 (44.5)
≥NPR 40,000	203 (55.5)
**Personal income**	
Yes	55 (15.0)
No	311 (85.0)
**Food security**	
Less than six months	21 (5.7)
Six or more months	345 (94.3)
**Alcohol consumption**	
No	323 (88.3)
Yes	43 (11.7)
**Tobacco use**	
No	275 (75.1)
Yes	91 (24.9)
**Chronic disease**	
No	139 (38.0)
Yes	227 (62.0)
**Family support**	
No	9 (2.5)
Yes	357 (97.5)
**Daily work support**	
No	50 (13.7)
Yes	316 (86.3)
**Emotional support**	
No	111 (30.3)
Yes	255 (69.7)
**Decisional support**	
No	106 (29.0)
Yes	260 (71.0)
**Economic support**	
No	75 (20.5)
Yes	291 (79.5)
**Accessibility**	
No accessibility	19 (5.2)
Accessibility	347 (94.8)
**Availability**	
No availability	204 (55.7)
Availability	162 (44.3)
**Affordability**	
No affordability	80 (21.9)
Affordability	286 (78.1)
**Regular physical activity**	
No	160 (43.7)
Yes	206 (56.3)

Others *: Buddhist/Muslim; others ^#^ daughter/alone/mother.

**Table 2 ijerph-22-00693-t002:** Summary of different domains of quality of life among older people of Central Nepal, 2023 (N = 366).

Domains of Quality of Life	Mean	SD	Minimum	Maximum
Sensory abilities	9.18	2.65	5.00	18.00
Autonomy	13.89	3.16	4.00	20.00
Past, present, and future activities	14.97	2.33	6.00	20.00
Social participation	14.29	2.17	7.00	20.00
Death and dying	7.35	4.21	4.00	20.00
Intimacy	14.69	2.36	8.00	20.00
**Overall QOL**	**74.38**	**7.82**	**51.00**	**103.00**

QOL: quality of life. SD: standard deviation.

**Table 3 ijerph-22-00693-t003:** Factors associated with overall quality of life among older people of Central Nepal, 2023.

Factors	β -Coefficients (95% Confidence Interval)	*p* Value
**Education**		<0.001
Literate (reference: illiterate)	3.771 (1.986, 5.556)	
**Household income**		0.02
NPR 40,000 and above (reference: NPR <40,000)	1.909 (0.337, 3.480)	
**Accessibility**		0.02
Accessible (reference: not accessible)	4.019 (0.666, 7.371)	
**Affordability**		<0.001
Affordable (reference: not affordable)	3.176 (1.327, 5.025)	
**Availability**		0.01
Availability (reference: no availability)	−2.011 (−3.585, −0.436)	
**Physical activity**		0.01
Yes (reference: No)	2.107 (0.607, 3.606)	

Variables entered in each model: gender, marital status, education, household income, food security, alcohol consumption, accessibility, availability, affordability, and physical activity. These variables were selected for inclusion in the model based on ANOVA test (see Appendix A). USD 1 = NPR 133.48 (for reference); NPR: Nepalese Rupees.

**Table 4 ijerph-22-00693-t004:** Factors associated with the individual domains of quality of life among older people of Central Nepal, 2023.

	Domains of QOL					
	β Coefficients (95% Confidence Interval) and *p*-Values					
Variables	Sensory Ability	Autonomy	Past, Present and Future Activities	Social Participation	Death and Dying	Intimacy
**Gender**	*p =* 0.73	***p* = 0.02**	*p* = 0.35	*p* = 0.55	***p* < 0.01**	*p* = 0.39
Female (reference: male)	0.12 (−0.57, 0.81)	−0.84 (−1.55, −0.12)	−0.25 (−0.78, 0.27)	−0.15 (−0.65, 0.35)	1.70 (0.62, 2.78)	−0.25 (−0.83, 0.33)
**Marital status**	*p* = 0.18	*p* = 0.32	*p* = 0.16	*p* = 0.08	*p* = 0.21	*p* = 0.82
Single (reference: married)	0.45 (−0.21, 1.12)	−0.35 (−1.03, 0.34)	−0.36 (−0.87, 0.14)	−0.43 (−0.92, 0.04)	−0.65 (−1.69, 0.38)	0.07 (−0.49, 0.62)
**Education**	*p* = 0.96	***p* = <0.001**	***p* = <0.01**	***p* = 0.01**	*p* = 0.50	***p* = 0.03**
Illiterate (reference: literate)	0.02 (−0.67, 0.70)	−1.23 (−1.94, −0.53)	−0.96 (−1.48, −0.44)	−0.61 (−1.11, −0.12)	−0.36 (−1.43, 0.70)	−0.62 (−1.19, 0.05)
**Household income**	*p* = 0.97	***p* = 0.04**	***p* = 0.02**	***p* = 0.01**	*p* = 0.21	***p* = 0.01**
NPR 40,000 and above (reference: NPR <40,000)	0.01 (−0.59, 0.61)	0.64 (0.01, 1.26)	0.55 (0.09, 1.00)	0.62 (0.18, 1.05)	−0.60 (−1.53, 0.34)	0.69 (0.19, 1.19)
**Food security**	*p* = 0.46	*p* = 0.838	***p* = 0.03**	*p* = 0.81	*p* = 0.25	*p* = 0.83
Six or more months (reference: less than six months)	−0.47 (−1.72, 0.78)	0.13 (−1.16, 1.43)	1.03 (0.08, 1.98)	0.11 (−0.80, 1.01)	−1.15 (−3.10, 0.80)	−0.11 (−1.16, 0.93)
**Alcohol consumption**	*p =* 0.31	***p* = 0.05**	*p* = 0.46	***p* = 0.02**	*p* = 0.87	*p* = 0.06
Yes (reference = no)	−0.46 (−1.35, 0.43)	0.93 (0.01, 1.85)	0.25 (−0.42, 0.93)	0.77 (0.12, 1.41)	−0.12 (−1.51, 1.27)	0.72 (−0.03, 1.46)
**Accessibility**	*p =* 0.22	*p =* 0.065	*p* = 0.410	*p* = 0.12	*p* = 0.89	*p* = 0.22
Accessibility (reference: no availability)	0.80 (−0.48, 2.10)	1.25 (−0.08, 2.58)	0.41 (−0.56, 1.38)	0.74 (−0.19, 1.67)	0.14 (−1.85, 2.14)	0.67 (−0.40, 1.74)
**Availability**	*p* = 0.63	***p* = 0.01**	***p* = <0.01**	*p* = 0.09	*p* = 0.12	***p* = 0.02**
Availability (reference: no availability)	0.15 (−0.45, 0.75)	−0.93 (−1.55, −0.30)	−0.88 (−1.33, −0.42)	−0.38 (−0.82, 0.06)	0.61 (−0.32, 1.55)	−0.59 (−1.09, −0.09)
**Affordability**	*p* = 0.29	***p* = 0.01**	***p* = <0.01**	***p* = <0.01**	*p* = 0.59	*p* = 0.10
Affordable (reference: not affordable)	−0.38 (−1.08, 0.33)	1.05 (0.32, 1.78)	1.09 (0.56, 1.63)	1.21 (0.70, 1.72)	−0.29 (−1.40, 0.80)	0.49 (−0.10, 1.08)
**Physical activity**	*p* = 0.43	***p* = 0.01**	*p* = 0.28	***p* = 0.01**	*p* = 0.63	*p* = 0.06
Yes (reference: no)	−0.23 (−0.80, 0.343)	0.84 (0.25, 1.44)	0.24 (−0.20, 0.67)	0.58 (0.16, 0.99)	0.22 (−0.67, 1.11)	0.46 (−0.02, 0.93)

Variables entered in each model: gender, marital status, education, household income, food security, alcohol consumption, accessibility, availability, affordability, and physical activity. These variables were selected for inclusion in the model based on ANOVA test (see Appendix A). USD 1 = NPR 133.48 (for reference); NPR: Nepalese Rupees.

## Data Availability

The raw data supporting the conclusions of this article will be made available by the corresponding author on reasonable request.

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
