# Peer review of "The Quality of Life and Associated Factors Among Older Adults in Central Nepal: A Cross-Sectional Study Using the WHOQOL-OLD Tool"

_ijerph, 2025, doi:10.3390/ijerph22050693_

Round 1
Reviewer 1 Report
Comments and Suggestions for Authors
Please see my summary, general, and detailed comments in the attached document.

Overall the quality of English is good, but the manuscript should be reviewed for consistency in tense (e.g., past or present), subject-verb agreement, and omission of sentence fragments. Some specific suggestions are noted in my review, but these suggestions are likely not exhaustive.
Reviewer 2 Report
Comments and Suggestions for Authors
The study appears to be well planned, performed and presented. My comments and suggestions are mostly on minor details.
Introduction, line 14: "..it is inevitable to ignore population phenomenon" This is a double negation and is better formulated as "..it is inevitable to pay attention to the population phenomenon" or maybe better "...one cannot ignore the population phenomenon"
Page 3: A series of language details:
Line 9: ..older person.. > ..older persons..
line 10: ...within older... > within the older ..
line 12: ..60 years and above.. > ...60 years or above...
line 13. ...of older person... > ..of the older person...
line 18: ...in the recent... > ..in recent...
Line 19: ...needs of older people into practice [22]... > ...needs of older people [22].
line 21: .. those HIV/AIDSs who... > ...those with HIV/AIDS who...
lines 22-23: Another study...........focused on... > Another study [25], conducted among older people attending an oupatient of a Kathmandu based hospital, focused on.....
line 26-27: ...status of general older... > ....status of the general older...
Line 27: ...of far western... > ...of the far western...
line 28: ..is -a socially.. > ... is socially,...
Page 4:
line 2: 3,69,377 should be 369,377 ?
Study design, line 5: ...and those were willing... >..and were willing...
line 7: ...communicate and those who were not willing to participate. A list.... >..communicate. A list... (the criterion of willing to participate is already stated two lines above)
Page 5: line 12: Explain briefly what is the meaning of 'food security'
Independent variables, second paragraph: According to my dictionary, accessibility and availabilty are synonyms. Please explain the difference between the two as used here. Any explanation for the surprisingly different for the two, with availability coming out as negatively related to QOL? Could it be a result of the statistical procedure, so that controling for one of the two similar domains artificially produced an inverse result of the other??
Page 6, line 4: 'de-identified' means 'anonymized' ?
Page 6-7: Referring values from Table 4: Be sure that the p-values given in the text are identical to those shown in the Table
Page 7 , heading of Table 1. Delete 'Central Nepal 2023'
Page 9, last lines of Table 1: There must be something between 'No physical activity at all' and 'Regular physical activity'. Should the first rather be 'No regular physical activity' ?
Page 10,Table 3: The coefficient and CI for education is given as '-0.415 (-5.556, -1.986)' . The coefficient should be within the CI, please correct
Page 12, Table 4: The association between Education and Intimacy is given as '-0.62 (-1.19, 0.05). As the 95% CI includes zero, p cannot be 0.03, as given. Please correct, either the CI or the p value.
Similarly, Table 4 : availability vs activities: CI includes zero, and p cannot be <0.01 as given. Correct
Page 16, supplementary Table 2: Instead of SD, I would prefer to have the SE, standard error of the mean, which relates more directly to the p values
Suppl. table 2: How could Age have a p=0.004 when the mean for both age groups are identical, 0,39 ?? Please correct
Reviewer 3 Report
Comments and Suggestions for Authors Overall Assessment This study demonstrates both novelty and validity in its investigation of quality of life (QOL) among the elderly in Nepal, with a unique focus on urban populations. The sample size is sufficient, and the results are statistically significant, lending high overall reliability to the findings. Comments However, there is a concern regarding the statistical analyses: the assumptions of normality and homogeneity of variances were not verified in advance. These checks are crucial as they can influence the test power and the interpretation of the results. Future research should address these aspects to further strengthen the study's methodological rigor.Author Response
See the attachment

Round 2
Reviewer 1 Report
Comments and Suggestions for Authors
The authors have made changes throughout the manuscript that have improved the clarity and quality of the paper considerably. The authors were responsive to my first suggestion (regarding the Introduction) and to my myriad smaller suggestions. I thank the authors for their time and efforts.
My second broad comment from my first review concerned the statistical choice to categorize quantitative/continuous variables for inclusion in the main analyses. As indicated in my previous review, this practice is highly discouraged among quantitative methodologists. I provided several citations from reputable publications (e.g. BMC, BMJ, Psychological Methods) to support my rationale and for the authors to read more about why this statistical practice is unnecessary at best and detrimental at worst. The authors responded with several citations in the domain of aging and quality of life which utilize this practice as justification for why they would not entertain changing their statistical decision. Although the authors' provided citations show the practice is common, it fails to prove the practice is recommended. Therefore, my suggestion that the authors utilize the original, quantitative format of their variables -- or to provide strong theory for the cutpoints used to categorize quantitative variables -- stands. Entertaining such a change does not require authors to collect additional data or drastically change their analytic framework. It is an easy adjustment to make that would improve the quality of the statistical analyses, which is in the best interest of the authors, IJERPH, and its readership.
Minor comments:
Table 4: Education now has two rows. I assume authors mean to delete the "illiterate" row; if so, please edit.
Author Response
My second broad comment from my first review concerned the statistical choice to categorize quantitative/continuous variables for inclusion in the main analyses. As indicated in my previous review, this practice is highly discouraged among quantitative methodologists. I provided several citations from reputable publications (e.g. BMC, BMJ, Psychological Methods) to support my rationale and for the authors to read more about why this statistical practice is unnecessary at best and detrimental at worst. The authors responded with several citations in the domain of aging and quality of life which utilize this practice as justification for why they would not entertain changing their statistical decision. Although the authors' provided citations show the practice is common, it fails to prove the practice is recommended. Therefore, my suggestion that the authors utilize the original, quantitative format of their variables -- or to provide strong theory for the cut points used to categorize quantitative variables -- stands. Entertaining such a change does not require authors to collect additional data or drastically change their analytic framework. It is an easy adjustment to make that would improve the quality of the statistical analyses, which is in the best interest of the authors, IJERPH, and its readership.
Author response: Thank you reviewer for their time and great feedback.This was also an eye opening discussion for us to start thinking about collecting as many continuous variables as possible. . We truly appreciate the insight. In the current study, the two variables collected as continuous were: age of the participants, and income. We have used median value as cut-off point. We ran additional analysis keeping these two variables as continuous variables. The results showed that the association of the variables remained the same and the age of the participant was no significant. We have added additional sentences in our result section. We performed a sensitivity analysis, treating participants’ age and income as continuous variables in the multiple regression model.
'While education, household income, accessibility, availability, affordability, and physical activity remained significantly associated with overall quality of life, respondents’ age did not show a significant association.'
Additionally, we also kept our existing analysis plus the results from sensitivity analysis and we hope this would be an acceptable approach to our reviewer.
Minor comments:
Table 4: Education now has two rows. I assume authors mean to delete the "illiterate" row; if so, please edit.
Author response: Thank you for identifying this. We have deleted the first row.